# Ranked from Within: Ranking Large Multimodal Models Without Labels

**Weijie Tu** [1]   **Weijian Deng** [1]   **Dylan Campbell** [1]   **Yu Yao** [2]   **Jiyang Zheng** [2]   **Tom Gedeon** [1 3 4]   **Tongliang Liu** [2]

## Abstract

Can the relative performance of a pre-trained large multimodal model (LMM) be predicted without access to labels? As LMMs proliferate, it becomes increasingly important to develop efficient ways to choose between them when faced with new data or tasks. The usual approach does the equivalent of giving the models an exam and marking them. We opt to avoid marking and the associated labor of determining the ground-truth answers. Instead, we explore other signals elicited and ascertain how well the models know their own limits, evaluating the effectiveness of these signals at unsupervised model ranking. We evaluate 47 state-of-the-art LMMs (*e.g.*, LLaVA) across 9 visual question answering benchmarks, analyzing how well uncertainty-based metrics can predict relative model performance. Our findings show that uncertainty scores derived from softmax distributions provide a robust and consistent basis for ranking models across various tasks. This facilitates the ranking of LMMs on unlabeled data, providing a practical approach for selecting models for diverse target domains without requiring manual annotation.

## 1. Introduction

Large multimodal models (LMMs), such as LLaVA (Liu et al., 2024c) and InstructBLIP (Dai et al., 2023), have demonstrated remarkable capabilities in handling a wide array of complex multimodal tasks, proving highly adaptable across diverse real-world applications (Liu et al., 2024a; Dai et al., 2023; Wang et al., 2024a; Huang et al., 2024; 2025; Zheng et al., 2025). From addressing scientific challenges (Lu et al., 2022; Hiippala et al., 2021) and performing optical character recognition (Mishra et al., 2019; Masry

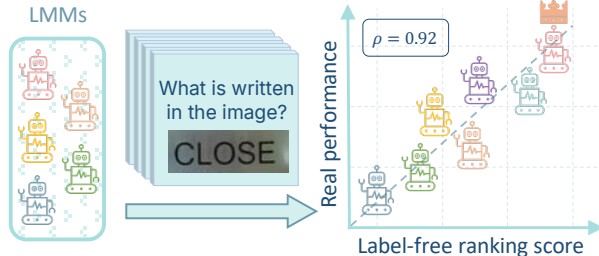

*Figure 1.* **Overview of unsupervised ranking for LMMs**. In scenarios where labeled data is scarce, selecting the best-performing model can be challenging. Our approach introduces a label-free proxy ranking score designed to reflect true performance, achieving a high correlation ($\rho = 0.92$) with actual metrics. This enables unsupervised comparison of LMMs, allowing users to identify the most suitable model without needing labeled data.

et al., 2022; Mathew et al., 2021) to identifying the spatial position of objects (Tong et al., 2024; x.ai, 2024), LMMs are increasingly widespread in practical settings. As LMMs proliferate, the need for rigorous evaluation metrics that accurately capture their capabilities and limitations has become more urgent. Thus, numerous benchmarks have been developed (Lu et al., 2022; Hiippala et al., 2021; Masry et al., 2022; Singh et al., 2019; Mathew et al., 2021; x.ai, 2024; Yue et al., 2024), aiming to provide reliable rankings and guide users in selecting models best suited for specific deployment scenarios.

However, these benchmarks are based on carefully curated datasets that can require substantial resources to develop and label. For many users who may not have access to these resources, assessing model performance can be challenging. Additionally, standard evaluations are often dataset-centric, depending on fixed, human-labeled metrics that may not capture the full range of model capabilities necessary for diverse applications. As LMM tasks continue to diversify and expand, evaluating them effectively has become increasingly complex, with new applications frequently needing additional data curation and specialized capabilities.

Addressing the challenge of efficiently evaluating a set of LMMs is important for their usability and effectiveness in deployment. As illustrated in Figure 1, selecting the most suitable LMM from a range of available options is

---

[1]Australian National University [2]Sydney AI Centre, The University of Sydney [3]Curtin University [4]University of ÓBuda. Correspondence to: Tom Gedeon <tom.gedeon@anu.edu.au>, Tongliang Liu <tongliang.liu@sydney.edu.au>.

*Proceedings of the 42$^{nd}$ International Conference on Machine Learning*, Vancouver, Canada. PMLR 267, 2025. Copyright 2025 by the author(s).

challenging in the absence of annotations. To address this, our work investigates label-free proxy ranking scores that closely align with the true performance ranking, enabling effective model comparison in label-scarce settings.

Our first finding is that the naïve approach of using the model's performance on an existing benchmark to rank its performance in a new target environment could be unstable. We further report that measures of prediction uncertainty are more effective ranking indicators. LMMs generate answers in an open-ended form by producing token sequences, with each token selected from the vocabulary based on the probability. This enables model uncertainty to be assessed using the logits at each token position.

To this end, we evaluate 45 different LMMs that have different training frameworks, *e.g.*, LLaVA-V1.5 (Liu et al., 2024a) and InstructBLIP (Dai et al., 2023), different visual encoders, *e.g.*, CLIP (Radford et al., 2021) and SigLIP (Zhai et al., 2023), and language models, *e.g.*, Vicuna (Team, 2023) and LLaMA (Touvron et al., 2023). We investigate three categories of model uncertainty approaches: softmax probabilities, self-consistency, and labeled proxy sets. We evaluate the ranking performance on 9 widely-adopted multimodal benchmarks, which span diverse domains, including reasoning scientific questions, recognizing optical characters, and identifying objects' spatial positions. Our main findings are that

- the performance of models on one dataset may not accurately reflect the ranking of the same models on a different dataset (Section 4);
- the effectiveness of ranking methods is influenced by task characteristics (*e.g.*, closed-form or free-form generation), but probability-based variants are typically quite robust and predictive (Section 5); and
- when examining correlations in model performance across different dataset pairs, we observe that text prompt similarity better correlates with model performance across datasets than image feature similarity (Section 6).

## 2. Related Work

**Unsupervised Model Ranking.** The goal of this task is to rank and select a best performant models without the access to the data annotations of target environments. The research on this task can date back to Forster *et al*. (Forster, 2000), and was further investigated in various tasks: (1) outlier detection (Zhao et al., 2022; 2021); (2) image classification (Kotary et al., 2022; Tu et al., 2024a; Zohar et al., 2024; Baek et al., 2022; Miller et al., 2021; Shi et al., 2024a); (3) time series anomaly detection (Ying et al., 2020); (4) multivariate anomaly detection in manufacturing systems (Engbers & Freitag, 2024), *etc*.

We discuss the most relevant two lines of research as follows. Miller *et al*. (2021) introduce an accuracy-on-the-line (AoL) phenomenon where strong linear correlation between probit-scaled in-distribution (ID) accuracy and out-of-distribution (OOD) accuracy across a variety of ML models. This means ID accuracy serves as a good indicator of model performance for a target domain. Shi *et al*. (2024a) revisit the established concept of lowest common ancestor (LCA) distance, which measures the hierarchical distance between labels and predictions within a predefined class hierarchy. By the observed linear correlation between ID LCA distance and OOD accuracy, it is viable to select a model based LCA distance of predictions. This work differs from prior study that we focus on ranking LMMs. This group of models makes predictions in a significantly different way from conventional classification models, which also results in different evaluation protocols.

**Uncertainty Estimation for LLMs and LMMs.** Uncertainty estimation seeks to quantify an ML model's confidence in its predictions (Guo et al., 2017). Recent studies have been dedicated to exploring uncertainty estimation specifically for LLMs (Kuhn et al., 2023; Malinin & Gales, 2021; Xiao & Wang, 2021; Huang et al., 2023; Lin et al., 2023; Kadavath et al., 2022; Azaria & Mitchell, 2023; Gottesman & Geva, 2024). For instance, Xiao *et al*. (2021) utilize ensemble methods to evaluate uncertainty in natural language generation models. Malinin *et al*. (2021) similarly introduce a unified approach to uncertainty estimation, leveraging ensemble methods, for autoregressive structured prediction tasks. To deal with the challenge of capturing "semantic equivalence" in natural language, Kuhn *et al*. (2023) propose semantic entropy, a method that utilizes linguistic invariances derived from shared meanings. Additionally, internal states of LMMs can be leveraged for uncertainty quantification or error detection by training a classifier (Azaria & Mitchell, 2023; Gottesman & Geva, 2024). This paper does not propose a new way to estimate uncertainty. Instead, it offers the novel insight that the existing uncertainty in LMM-generated outputs effectively reflects their relative performance across benchmarks without manual labels.

**Evaluation & Benchmarking LMMs.** The rapid development of LMMs has greatly propelled advancements in multimodal models, showcasing significant improvements in their perception and reasoning abilities. This shift has rendered traditional benchmarks, which focus solely on isolated task performance (Karpathy & Fei-Fei, 2015; Antol et al., 2015). Researchers have introduced new benchmarks to evaluate LMM in a broad spectrum of multimodal tasks (Goyal et al., 2017; Lin et al., 2014; Russakovsky et al., 2015). Recent studies (Yue et al., 2024; x.ai, 2024; Fu et al., 2023) highlight the need for more comprehensive benchmarks to effectively evaluate the reasoning and

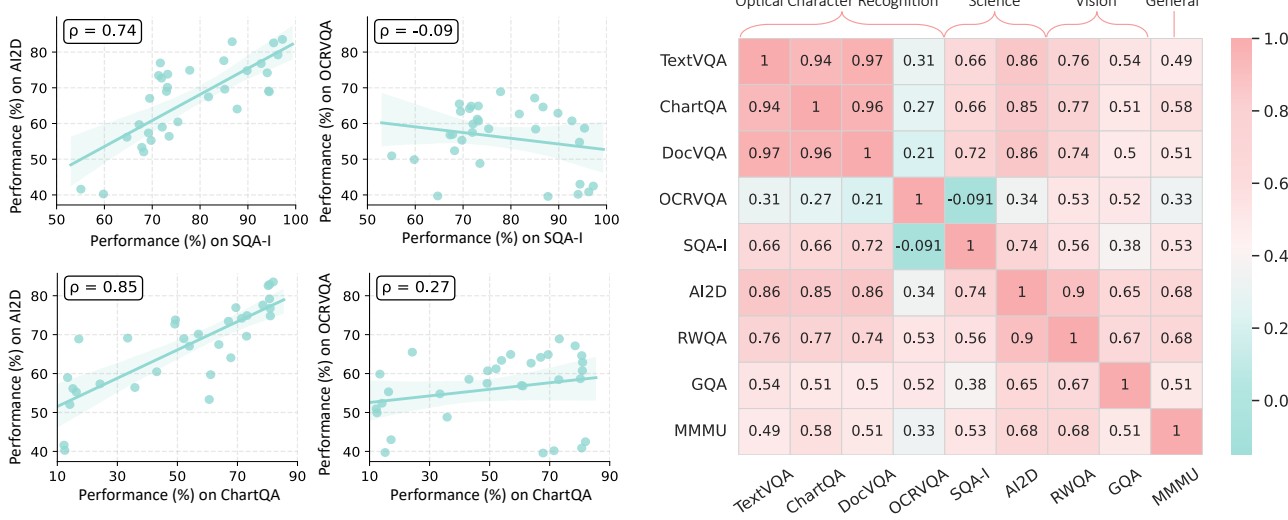

(a) Correlation study on model performance between benchmarks

(b) Correlation matrix for 9 benchmarks

*Figure 2.* **Correlation analysis of model performance** across benchmarks. **(a)** Scatter plots illustrating the Spearman's rank correlation coefficients ($\rho$) between performance on selected benchmarks, indicating how well performance on one benchmark predicts performance on another. Each point represents a model. The straight lines are fit with robust linear regression (Huber, 2011). **(b)** Heatmap of the correlation matrix for performance across eight benchmarks, with color intensity representing the strength of correlation. Higher correlations (closer to 1) appear in red, while weaker correlations approach blue. The varying correlation strength indicates that using performance on one benchmark to rank LMMs in a target deployment environment may be inconsistent or unreliable.

understanding abilities of LMMs. For instance,

Several benchmarks (x.ai, 2024; Ainslie et al., 2023; Tong et al., 2024) have been developed to assess multimodal models' real-world spatial understanding. Lu *et al*. (2024) and Zhang *et al*. (2024) introduce benchmarks specifically designed to evaluate MLLMs' mathematical reasoning, focusing on their ability to comprehend and reason about visual mathematical figures. Yue *et al*. (2024) carefully curate a diverse set of multi-discipline tasks that require college-level subject knowledge and complex reasoning. Additionally, numerous benchmarks (Mishra et al., 2019; Masry et al., 2022; Mathew et al., 2021; Liu et al., 2023) assess the performance of LMMs in optical character recognition.

## 3. Task Formulation

**Task Definition.** Let a multimodal task be represented by a dataset $T = \{\mathbf{x}_i, \mathbf{y}_i\}_{i=1}^{N}$, where $\mathbf{x}_i$ and $\mathbf{y}_i$ denote the input prompt and the corresponding answer for the $i$-th sample, respectively. We have access to $M$ large multimodal models (LMMs), denoted as $\{f_m\}_{m=1}^{M}$. Each LMM $f_m$, with pre-trained weights $\boldsymbol{\theta}_m$, generates a sequence of tokens $\{z_k\}_{k=1}^{K}$ from the input prompt $\mathbf{x}$ via a decoding process: $z_k = f_m([\mathbf{x}, z_1, z_2, \ldots, z_{k-1}] \mid \boldsymbol{\theta}_m)$, where $z_k$ represents the $k$-th generated token. To assess the performance of each LMM, this task employs an evaluation metric (*e.g.*, accuracy) that determines the ground-truth performance $\{g_m\}_{m=1}^{M}$ by comparing the generated sequences with the

ground-truth answers $\mathbf{y}_i$.

The objective of this paper is to develop methods for computing a score $s_m$ for each LMM without requiring access to task-specific data annotations. Ideally, these computed scores should closely correlate with ground-truth performance, allowing us to rank and select LMMs based on their performance using only these scores.

**Evaluation Metric.** We use Spearman's rank correlation coefficient $\rho$ (Kendall, 1948) to evaluate the monotonic relationship between scores and model performance. Additionally, we calculate Kendall's weighted rank correlation $\tau_w$ (Shieh, 1998), which effectively highlights top-ranked items (You et al., 2021). Both coefficients range from $[-1, 1]$, with values near $-1$ or $1$ indicating strong negative or positive correlations, and $0$ indicating no correlation.

## 4. Uncertainty for Ranking LMMs

This section discusses the unique characteristics of ranking various LMMs compared to conventional ML models. We then introduce three distinct approaches that leverage uncertainty in model predictions for ranking.

### 4.1. What Makes Ranking LMMs Interesting?

**Unique Challenges for Ranking LMMs.** While LMMs can be considered a subset of machine learning (ML) mod-

els, ranking them introduces unique challenges not present in traditional ML models. Below, we discuss the unique characteristics of LMMs and the challenges they bring to the model-ranking process. First, LMMs often have billions of pre-trained parameters, which presents significant challenges for traditional "white-box" analysis for ranking various LMMs. Second, these models are typically trained on large datasets of instruction fine-tuning samples, which may be proprietary data. As a result, risk assessment methods that require access to training data are either unsuitable or should be adjusted. Additionally, although some LMMs may disclose their final model weights, other information, such as training loss and intermediate checkpoints, often remains undisclosed. The lack of the access to training details limits the use of techniques in unsupervised accuracy estimation (Deng & Zheng, 2021; Tu et al., 2023).

**Does Accuracy-on-the-Line (AoL) Suffice?** Given these challenges, one may consider leveraging the AoL phenomenon (Miller et al., 2021) as a potential method for ranking LMMs. AoL refers to the strong linear correlation between probit-scaled in-distribution (ID) accuracy and out-of-distribution (OOD) accuracy across various ML models. This suggests that ID accuracy could serve as a reliable predictor of OOD performance, making AoL suitable for selecting LMMs in target testing environments. However, there are several reasons why this is not the case. First, obtaining ID performance data is often impractical, as LMMs are typically trained on large, sometimes proprietary datasets, limiting direct access to ID metrics. Second, while performance on existing benchmarks is more readily available, using this data to rank models for new deployment environments is problematic. Figure 2 illustrates the correlation between proxy benchmark performance and target testing environments, revealing extreme variability in correlation strength. This highlights the unreliability of using benchmark performance as the sole ranking criterion. Third, relying solely on proxy benchmark performance fails to capture the unique statistical characteristics of target testing datasets. AoL tends to rank models identically across different environments, regardless of the actual deployment context.

**What can we use for ranking LMMs?** Our approach leverages the readily available outputs of LMMs—specifically, token prediction logits and generated tokens—without requiring intricate architectural analysis or complex extraction techniques. Inspired by recent work on uncertainty scores for classifier ranking (Hu et al., 2024; Tu et al., 2024b), we introduce a novel adaptation of these techniques tailored to the specific characteristics of LMMs. By analyzing the distribution of prediction logits and the variability in generated outputs, we aim to assess each model's self-awareness of its limitations. This enables an unsupervised model ranking method, focusing primarily on tech-

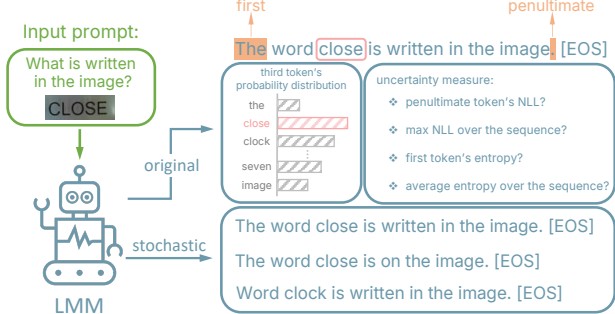

*Figure 3.* **An example of running one LMM for a VQA task.** We also present different token positions, methods to compute token-level uncertainty and the generation of stochastic predictions.

niques that utilize these readily accessible LMM outputs.

## 4.2. Assessing Uncertainty by Softmax Probabilities

LMMs generate answers in an open-form manner by producing sequences of tokens. The selection of each token can be viewed as a classification process, where the model selects the token with, *e.g.*, the highest probability over its entire vocabulary. This allows us to assess model uncertainty based on the logits at each token position.

For token-level uncertainty (Huang et al., 2023), we can focus on two specific positions: the first token and the penultimate token (*i.e.*, the token preceding the end-of-sequence token), as illustrated in Figure 3. The logit of the first token reflects the model's initial response to the input prompt, while the logit of the penultimate token captures the model's understanding of both the prompt and the generated response. The uncertainty associated with these two positions may provide insight into the model's overall confidence in answering the question. Beyond token-level uncertainty, sentence-level uncertainty can be calculated by aggregating uncertainty across all tokens in the sequence (Manakul et al., 2023; Huang et al., 2023). Specifically, sentence-level uncertainty can be quantified using the mean or the maximum negative log-likelihood (NLL) values across the entire generated sequence:

$$\mathbf{NLL}_{\max} = \max_{j} \left( -\log p_{ij} \right) \quad (1)$$

$$\mathbf{NLL}_{\text{avg}} = -\frac{1}{J} \sum_{j=1}^{J} \log p_{ij} \quad (2)$$

where $p_{ij}$ is the likelihood of the word generated by the LMM at the $j$-th token of the $i$-th sentence and $J$ is the number of tokens generated in the sentence. Equation (1) quantifies a sentence's uncertainty via the least likely token, while Equation (2) uses the average per-token log-likelihood, allowing for length-independent comparisons

of uncertainty (Malinin & Gales, 2020; Murray & Chiang, 2018). Moreover, Equation (2) relates to perplexity, $\exp \text{Avg}(-\log p)$ (Jelinek et al., 1977; Manning & Schütze, 1999), a standard measure of model quality.

Alternatively, entropy $\mathcal{H}$ can be used instead of the negative log-likelihood to assess uncertainty. In the context of LMMs ranking, we adopt normalized entropy to account for varying vocabulary sizes across models, thus scaling entropy to the interval $[0, 1]$ and making it comparable across different models. The normalized entropy is given by:

$$\mathcal{H}_{ij} = -\frac{1}{\log |\mathcal{W}|} \sum_{w \in \mathcal{W}} p_{ij}(w) \log p_{ij}(w) \qquad (3)$$

where $p_{ij}(w)$ is the likelihood of the word $w$ being generated at the $j$-th token of the $i$-th sentence, and $W$ is the set of all possible words in the vocabulary.

There are eight variants of output probability-based methods, denoted as $\textbf{NLL}_\text{F}$, $\textbf{NLL}_\text{P}$, $\textbf{NLL}_\text{max}$, $\textbf{NLL}_\text{avg}$, $\textbf{Ent}_\text{F}$, $\textbf{Ent}_\text{P}$, $\textbf{Ent}_\text{max}$, and $\textbf{Ent}_\text{avg}$. "NLL" and "Ent" represent negative log-likelihood and entropy, respectively, while "F" and "P" refer to the first and penultimate tokens.

### 4.3. Assessing Uncertainty by Self-Consistency

Another approach involves examining the non-deterministic generations produced by models. The core intuition is that a more accurate model will produce predictions closely aligned with the original answer, while a less accurate model may yield more divergent responses with each inference. In the case of LMMs, the temperature parameter $t$ controls the randomness of predictions: a temperature of zero forces deterministic predictions, where the model selects only the token with the highest probability. When $t$ is larger than $0$, the model generates tokens stochastically, selecting tokens based on probabilities above a threshold. As $t$ increases, tokens are sampled from an increasingly uniform distribution (Chen et al., 2023b; Cobbe et al., 2021).

To analyze the consistency in these stochastic predictions, we explore two common methods: BLEU (Papineni et al., 2002) and BERTScore (Zhang et al., 2019). BLEU is a n-gram-based metric that evaluates the similarity of generated sequences to the reference answer, while BERTScore uses a pre-trained language model to embed answers and measures similarity in embedding space. Then, we use the mean value of similarities to represent the consistency for the input sample, which can be denoted as $\frac{1}{T} \sum_{i=1}^{T} \text{sim}(P_i, P_\text{ori})$, where $T$ is the number of stochastic inferences, $\text{sim}(\cdot)$ is the similarity function (*e.g.*, BLEU), and $P_i$ and $P_\text{ori}$ are the $i$-th stochastic prediction and the original answer, respectively. Following the practice outlined in (Chen et al., 2023b; Cobbe et al., 2021; Huang et al., 2023), we collect five stochastic inferences per sample and set $t = 0.7$ to maintain a relatively high degree of stochasticity in LMM

generation while keeping compute overhead manageable. We adopt a unigram BLEU and denote the methods using BLEU and BERTScore as $\textbf{Sample}_\text{BLEU}$ and $\textbf{Sample}_\text{BERT}$, respectively. Furthermore, BERTScore tends to assign high similarity scores between single letters, such as "A" and "B", which should be considered very different in the contexts of MCVQ. For example, BERTScore gives a high similarity score of $0.998$ for "A" and "B", making it ineffective for distinguishing model performance. To address this, we modified the response to include the full answer text (*e.g.*, "A. North America"), denoted as $\textbf{Sample}^*_\text{BERT}$.

### 4.4. Assessing Uncertainty With Labeled Proxy Data

Beyond the uncertainty score calculated directly on the target dataset, we also explore scores obtained using a proxy labeled dataset. Garg et al. (2022) proposed average thresholded confidence (ATC), which calculates a threshold $\delta$ on a validation set (*e.g.*, CIFAR-10 (Krizhevsky & Hinton, 2009) validation set) and considers an image correctly classified on a new dataset with the same task if its confidence score exceeds the threshold. With the derived threshold, model performance on the target domain is estimated by the proportion of samples with confidence scores higher than the threshold. For LMMs, we use an existing benchmark as a proxy set to calculate $\delta$. The calculation of ATC is

$$\text{ATC} = \mathbb{E}_{x \in T}[\mathbb{I}[u(f(x)) > \delta]], \qquad (4)$$

where $\mathbb{I}$ is a binary indicator function, and $u(\cdot)$ represents an uncertainty estimation method, for which we use $\textbf{NLL}_\text{max}$. ATC provides both a ranking of model performance and an estimate of the expected performance.

## 5. Experiments

In this section, we first introduce the experimental setup including the evaluated datasets and models we considered. Then, we show the results of ranking different large multimodal models (LMMs).

### 5.1. Experiment Setup

**Benchmarks.** We choose the task of visual question answering, which is a common way to evaluate LMMs (Liu et al., 2024a; Dai et al., 2023; Beyer et al., 2024; Lu et al., 2022; Yue et al., 2024). We evaluate on multiple choice visual question (MCVQ) and visual question answering (VQA) benchmarks. MCQ and VQA are both types of question-answering formats for evaluating LMMs. For the former, the model is provided with several answer options, out of which the correct subset is to be selected. In contrast, the latter is usually open-ended and the models may generate answers freely. We consider $8$ widely-adopted MCVQ and VQA benchmarks. They are (1) the

| Method | MCVQ | | | | | | | | VQA | | | | | | | | | | Average | |
|---|---|---|---|---|---|---|---|---|---|---|---|---|---|---|---|---|---|---|---|---|
| | SQA-I | | AI2D | | RWQA | | MMMU | | GQA | | ChartQA | | OCRVQA | | TextVQA | | DocVQA | | | |
| | $\rho$ | $\tau_w$ | $\rho$ | $\tau_w$ | $\rho$ | $\tau_w$ | $\rho$ | $\tau_w$ | $\rho$ | $\tau_w$ | $\rho$ | $\tau_w$ | $\rho$ | $\tau_w$ | $\rho$ | $\tau_w$ | $\rho$ | $\tau_w$ | $\rho$ | $\tau_w$ |
| AoL | 0.52 | 0.45 | 0.73 | 0.61 | 0.70 | 0.57 | 0.54 | 0.45 | 0.53 | 0.32 | 0.69 | 0.59 | 0.30 | 0.20 | 0.69 | 0.57 | 0.68 | 0.60 | 0.60 | 0.48 |
| $NLL_F$ | 0.83 | 0.78 | 0.84 | 0.76 | 0.56 | 0.58 | 0.60 | 0.59 | 0.71 | 0.57 | 0.63 | 0.41 | 0.78 | 0.64 | 0.74 | 0.63 | 0.84 | 0.74 | 0.73 | 0.63 |
| $NLL_P$ | 0.64 | 0.60 | 0.61 | 0.58 | 0.25 | 0.37 | 0.16 | 0.18 | 0.58 | 0.52 | 0.79 | 0.67 | 0.81 | 0.68 | 0.71 | 0.67 | 0.88 | 0.78 | 0.60 | 0.56 |
| $NLL_{max}$ | 0.80 | 0.76 | 0.91 | 0.81 | 0.63 | 0.63 | 0.49 | 0.44 | 0.72 | 0.59 | 0.94 | 0.80 | 0.64 | 0.63 | 0.83 | 0.69 | 0.92 | 0.82 | 0.76 | 0.69 |
| $NLL_{avg}$ | 0.72 | 0.64 | 0.88 | 0.73 | 0.64 | 0.56 | 0.50 | 0.40 | 0.67 | 0.55 | 0.92 | 0.75 | 0.81 | 0.65 | 0.81 | 0.72 | 0.93 | 0.82 | 0.76 | 0.65 |
| $Ent_F$ | 0.59 | 0.51 | 0.82 | 0.59 | 0.65 | 0.56 | 0.64 | 0.52 | 0.54 | 0.20 | 0.64 | 0.45 | 0.69 | 0.57 | 0.71 | 0.56 | 0.80 | 0.70 | 0.68 | 0.52 |
| $Ent_P$ | 0.49 | 0.39 | 0.69 | 0.48 | 0.43 | 0.43 | 0.34 | 0.24 | 0.46 | 0.21 | 0.82 | 0.64 | 0.80 | 0.64 | 0.70 | 0.54 | 0.86 | 0.68 | 0.62 | 0.47 |
| $Ent_{max}$ | 0.58 | 0.39 | 0.82 | 0.59 | 0.67 | 0.60 | 0.66 | 0.54 | 0.54 | 0.21 | 0.88 | 0.66 | 0.53 | 0.52 | 0.76 | 0.60 | 0.87 | 0.73 | 0.70 | 0.54 |
| $Ent_{avg}$ | 0.58 | 0.33 | 0.80 | 0.56 | 0.62 | 0.50 | 0.59 | 0.41 | 0.57 | 0.26 | 0.91 | 0.68 | 0.77 | 0.62 | 0.74 | 0.58 | 0.88 | 0.74 | 0.72 | 0.52 |
| $Sample_{BLEU}$ | 0.65 | 0.68 | 0.76 | 0.59 | 0.48 | 0.36 | 0.37 | 0.12 | 0.44 | 0.41 | 0.89 | 0.61 | 0.47 | 0.58 | 0.81 | 0.60 | 0.90 | 0.63 | 0.64 | 0.51 |
| $Sample_{BERT}$ | 0.46 | 0.52 | 0.70 | 0.55 | 0.52 | 0.28 | 0.45 | 0.44 | 0.51 | 0.47 | 0.90 | 0.61 | 0.75 | 0.72 | 0.77 | 0.59 | 0.89 | 0.64 | 0.66 | 0.54 |
| $Sample^*_{BERT}$ | 0.67 | 0.56 | 0.78 | 0.62 | 0.67 | 0.48 | 0.59 | 0.39 | 0.51 | 0.47 | 0.90 | 0.61 | 0.75 | 0.72 | 0.77 | 0.59 | 0.89 | 0.64 | 0.73 | 0.56 |
| ATC | 0.73 | 0.74 | 0.80 | 0.72 | 0.51 | 0.40 | 0.41 | 0.35 | 0.39 | 0.20 | 0.95 | 0.85 | 0.28 | 0.21 | 0.69 | 0.64 | 0.89 | 0.77 | 0.63 | 0.54 |

*Table 1.* **Method comparison across eight multimodal tasks**. The table presents a comparison of four groups of methods: accuracy-based, output-probability-based, sample-based, and unsupervised model evaluation methods. We evaluate these methods using Spearman's rank correlation ($\rho$) and weighted Kendall's correlation ($\tau_w$). Both coefficients range from $-1$ to $1$, where values close to $-1$ or $1$ indicate strong negative or positive correlations, respectively, and $0$ indicates no correlation. The **AoL** and **ATC** performance is calculated as the average correlation when using the scores computed on the other seven domains and the model performance on the target domain. The highest correlation values for each task are highlighted in green, while the second highest values are marked in blue. All methods are ranked to **three** decimal places. Note that, we use the absolute value of correlation strength in the table for **NLL** and **Ent**. The results indicate that $NLL_{max}$ and $NLL_{avg}$ are often preferable, as they show greater stability and stronger correlation with model performance.

subset of ScienceQA (Lu et al., 2022) with images (SQA-I) and AI2D (Hiippala et al., 2021) which assess LMMs' scientific knowledge; (2) ChartQA (Masry et al., 2022), OCRVQA (Mishra et al., 2019), TextVQA (Singh et al., 2019) and DocVQA (Mathew et al., 2021) to examine their ability to recognize optical character; (3) RealWorldQA (RWQA) (x.ai, 2024) and GQA (Ainslie et al., 2023) which evaluate LMMs' vision-centric capability; (4) MMMU (Yue et al., 2024) which assays LMMs on multi-disciplinary tasks that demand college-level subject knowledge. Note that SQA-I, AI2D, RWQA and MMMU are MCVQ datasets, while the others are VQA.

**Models.** The goal is to choose the best LMM over all different series of LMMs. We include LLaVA-V1.5 (Liu et al., 2024a), ShareGPT4V (Chen et al., 2023a), LLaVA-NeXT (Liu et al., 2024b), InstructBLIP (Dai et al., 2023), LLaVA-NeXT-Interleave (Li et al., 2024b), LLaVA-OneVision (Li et al., 2024a), Eagle (Shi et al., 2024b), mPLUG-Owl (Li et al., 2022), InternVL (Chen et al., 2024), PaliGemma (Beyer et al., 2024), Mantis (Jiang et al., 2024), DeepSeek-VL2 (Wu et al., 2024) and Qwen2-VL (Wang et al., 2024b). In total, we collect 32 different models, which all can be accessed on Hugging Face (Wolf et al., 2020).

### 5.2. Key Findings

**Accuracy-on-the-Line is unreliable for ranking LMMs in new domains.** Table 1 summarizes the ranking capability of all methods across eight benchmarks. The AoL performance is calculated as the average correlation strength when using the other seven domains to predict model rankings in the target domain. We observe that AoL does not achieve

consistently high correlation with model performance in seven out of the eight benchmarks. Although AoL shows strong results on RealWorldQA, where it performs best, uncertainty-based methods also demonstrate high correlation strength. For instance, $NLL_{max}$ exhibits a Spearman's $\rho$ 0.07 lower but a weighted Kendall's $\tau_w$ 0.06 higher than AoL. These findings suggest that relying on existing benchmarks alone to select models for deployment could be risky and unstable, as they may not well capture the statistical characteristics of the new target domain.

**The choice of token matters for output probability-based ranking.** We studied four ways of using tokens for estimating model uncertainty. For the two on specific positions, the first and penultimate tokens, their predictive performance depends on the task type (*i.e.*, MCVQ *vs.* VQA). The uncertainty associated with generating the first token is more indicative for MCVQ tasks, while the penultimate token generally proves more predictive for VQA tasks. This difference is due to the nature of MCVQ tasks, which often prompt models to generate only a single option letter (*e.g.*, "A"), making the first token crucial for evaluating response accuracy. In contrast, VQA tasks require open-form responses consisting of multiple tokens, making the penultimate token—reflecting the model's understanding of both the question and the answer—more informative.

For the two that consider every token in the generated output, *i.e.*, $NLL_{max}$ and $NLL_{avg}$, we find that they are more stable and less task-specific compared to variants that consider individual tokens. While $NLL_{max}$ and $Ent_{max}$ (reflecting the least confident token) is more effective for MCVQ tasks, both methods perform similarly on VQA tasks. The higher

ranking correlation of using the first token and the least confident token as ranking indicators for MCVQ suggests that a single token (typically the option letter) holds significant meaning. Considering all potential tokens can sometimes result in an unexpectedly higher confidence level, as LMMs may generate the complete answer alongside the option letter (*e.g.*, "B. Columbia"). The tokens following the option letter often have high confidence levels, leading to an overall increase in estimated uncertainty. This tendency results in models generating entire answers with lower uncertainty scores, yielding higher ranks. These findings underscore the importance of further exploration into the optimal token positions for uncertainty estimation in LMMs. One potential approach is leveraging a language model to identify the position of the option letter and use its softmax probability as the uncertainty measure for the complete response.

**The negative log-likelihood (NLL) is more stable and predictive for LMMs ranking than normalized entropy.** For seven out of eight benchmarks, NLL consistently shows stronger correlation with model performance than entropy. Both NLL and entropy make use of the softmax probability distribution predicted for each token during generation. However, the NLL-based approaches only consider the maximum probability in this distribution, while the entropy considers all entries. The lower correlation strength of normalized entropy may therefore stem from the significantly different vocabulary sizes of various LMMs. For instance, LLaVA-V1.5-7B has a vocabulary of 32k tokens, while PaliGemma-3B-mix has a much larger vocabulary of 257k tokens. Despite being normalized by vocabulary size, entropy may still be more susceptible to noise than NLL. Future work could include applying dimension reduction techniques or limiting consideration to the top-$k$ most probable tokens or a pre-defined set of tokens.

**Sample-based methods are strong candidates for model ranking without intrinsic access to LMMs.** The correlation strength of sample-based methods is influenced by the nature of the task. They yield higher correlation scores on VQA tasks compared to MCVQ tasks. In VQA, models typically produce more varied responses, making BLEU and BERTScore effective for capturing uncertainty. However, MCVQ tasks constrain models to select from a pre-defined set of answers. While non-zero temperature introduces some randomness, the variation between stochastic inferences is limited. Additionally, we also find that $\mathbf{Sample}^{*}_{\mathrm{BERT}}$ has higher correlation scores than $\mathbf{Sample}_{\mathrm{BERT}}$ on four MCVQ benchmarks and suggests that developing more advanced algorithms to capture uncertainty from multiple stochastic inferences could be beneficial.

Although sample-based methods show weaker overall correlations, they remain competitive without requiring access to

| Method | MCVQ | | VQA | | Average |
| --- | --- | --- | --- | --- | --- |
| | AI2D | MMMU | TextVQA | ChartQA | |
| **AoL** | 0.42 | 0.43 | 0.57 | 0.64 | 0.52 |
| $\mathbf{NLL}_{\mathrm{max}}$ | 0.29 | 0.59 | 0.97 | 0.99 | 0.71 |
| $\mathbf{NLL}_{\mathrm{avg}}$ | 0.26 | 0.28 | 0.98 | 0.98 | 0.63 |
| $\mathbf{Ent}_{\mathrm{max}}$ | 0.37 | 0.62 | 0.97 | 0.96 | 0.73 |
| $\mathbf{Ent}_{\mathrm{avg}}$ | 0.11 | 0.38 | 0.98 | 0.98 | 0.63 |
| $\mathbf{Sample}_{\mathrm{BLEU}}$ | 0.63 | 0.73 | 0.76 | 0.64 | 0.69 |
| **ATC** | 0.12 | 0.59 | 0.85 | 0.84 | 0.60 |

*Table 2.* **Method comparison for ranking different LLaVA-prismatic** models on AI2D and TextVQA. We use Spearman's rank correlation ($\rho$) as the metric. We observe that uncertainty-based method still exhibit moderately high correlation strength, which indicate their effectiveness in ranking LLaVA models.

model architecture or internal states. This highlights their potential to rank API-based or closed-source LMMs (*e.g.*, GPT-4V (Achiam et al., 2023)).

**ATC can be used for LMMs ranking, but the correlation strength is influenced by the choice of proxy dataset.** We compute ATC performance as the average correlation when using the other seven datasets as proxy datasets. Our analysis reveals the choice of proxy dataset is critical, as the scale of uncertainty calculated on different datasets can vary. This variation can lead to an inaccurately estimated threshold for determining instance correctness. A potential improvement involves adopting uncertainty calibration methods, such as temperature scaling (Guo et al., 2017), to calibrate model uncertainty onto a consistent scale.

## 6. Analysis

This section includes three distinct analyses. The first specifically investigates whether the considered methods are effective for ranking models within the same series. We use LLaVA models as a case study because they are widely adopted and representative of LMM architectures. The other two analyses consider models from different series, consistent with the broader evaluation in Section 5.

**Ranking LMMs from the same series.** So far, all experiments have focused on selecting LMMs from different model series. However, in some scenarios, the objective is to choose a training recipe that yields better-performing LMMs within the same model series. These models may be trained with additional fine-tuning steps, varied data augmentations, different training sources, or different language models (*e.g.*, Vicuna and Mistral (Jiang et al., 2023)) and visual encoders, such as CLIP and DINOv2 (Oquab et al., 2023). For our analysis, we use the LLaVA series due to its widespread adoption. Specifically, we employ 15 different LLaVA prismatic (LLaVA-pri) models (Karamcheti

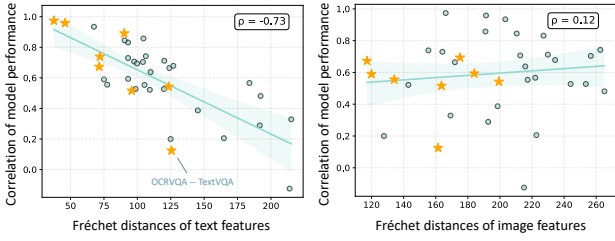

*Figure 4.* **Correlation Analysis of Fréchet Distances and Model Performance Correlation Across Datasets.** Orange stars indicate the dataset pairs with the highest similarity for each dataset. Observations reveal that variations in text prompt similarity are more closely aligned with changes in performance correlation than variations in image feature similarity.

et al., 2024). We report results on four datasets in Table 2: AI2D, MMMU, TextVQA, and ChartQA, with AI2D and TextVQA being used to evaluate the performance of different LLaVA variants in the original paper. The **AoL** and **ATC** performance metrics are computed in the same manner as described in Section 5, using the average correlation across other three datasets.

Our findings align with those from ranking different series of LMMs. First, using model performance on a single existing dataset may not accurately reflect LMMs rankings on a different domain, even when the models are trained with a similar pipeline and minor variations. Second, we observe that ranking different LLaVA-pri models in MCVQ presents a significant challenge, as the variance in model performance on MCVQ tasks is lower compared to VQA tasks. For instance, the performance gap on AI2D is only $4\%$, whereas the gap on TextVQA exceeds $20\%$. This indicates that ranking methods must capture subtle differences between models. We find that **Sample**$_{\text{BLEU}}$ remains effective, while NLL and Entropy may not capture these nuances accurately. Additionally, the decrease in correlation between uncertainty estimated by softmax probability suggests that although modifications to the training pipeline may have a slight effect on performance, they can lead to significant variations in the confidence levels for generation. This finding underscores the importance of assessing LMMs more comprehensively, beyond accuracy alone.

**Analysis of the weak correlation of AoL.** Figure 2(b) illustrates the correlation of model performance across different benchmarks. We observe that while TextVQA, ChartQA, DocVQA, and OCRVQA all aim to assess the capability of LMMs to recognize optical characters, the correlation between them varies significantly. Specifically, model performance on TextVQA, ChartQA, and DocVQA shows strong correlations, whereas performance on OCRVQA consistently exhibits low correlation with the other three datasets. To explore whether the images or texts within these datasets

| Method | MCVQ | | VQA | |
|---|---|---|---|---|
| | AI2D | MMMU | TextVQA | ChartQA |
| **#Samples** | 3088 | 1050 | 5000 | 2500 |
| **50 samples** | 0.89 | 0.35 | 0.92 | 0.98 |
| **NLL**$_{\text{max}}$ | 0.92 | 0.56 | 0.82 | 0.93 |

*Table 3.* **Labelling a subset of target domain** to rank models on AI2D, MMMU, TextVQA and ChartQA. We report the Spearman's rank correlation ($\rho$). While a small labeled set provides a reasonable ranking, it may not fully capture the overall order. In contrast, NLL$_{\text{max}}$ offers more stable correlations, highlighting its potential for label-free model ranking.

influence the correlation strength of model performance, we utilize CLIP-L-14-336 (Radford et al., 2021) to extract image and text embeddings. We then use the Fréchet distance (FD) (Fréchet, 1957) to measure the similarity between datasets based on these features.

Figure 4 presents a correlation study between the FD of dataset pairs and the correlation strength of model performance on those datasets. We observe a strong correlation for FDs computed using text input features, while FDs measured by image features show a weaker correlation. This suggests that text input similarity is likely a more influential factor for model performance correlation than image similarity. Additionally, orange stars are used to label the points representing the lowest FD for each dataset. Moreover, OCRVQA shows a low correlation strength with other datasets (Figure 2). The closest dataset in prompt feature space is TextVQA, with a FD of 124, which is considerably higher than the lowest FDs observed between other dataset pairs, typically around 70 or lower. This difference sheds light on the weak model performance correlation between OCRVQA and other OCR-focused datasets.

**Ranking LMMs by a labeled subset of the target domain.** Table 3 presents the correlation strength between model performance on 50 labeled instances in the target domain and the overall performance across the entire dataset. We observe that a small labeled set can provide a good ranking. However, such an approach may not fully capture the model ranking across the entire dataset, since the sampled data may not be representative (Polo et al., 2024). In contrast, **NLL**$_{\text{max}}$ gives a more stable correlation, highlighting the potential of uncertainty-based methods for effective model ranking without data annotation.

## 7. Conclusion and Discussion

This work studied whether the performance of large multimodal models in a new target domain can be ranked without the access to target domain labels. Our analysis identified only a weak correlation in model performance across

different domains. This motivated the investigation and evaluation of alternative approaches based on uncertainty estimates obtained from model predictions. We evaluate 45 LMMs on closed and open-ended visual question answering tasks, testing various training frameworks, visual encoders, and language models. Our experiments reveal that scores based on the negative log-likelihood of generated tokens serve as highly effective performance indicators for target domains. We also find that while stochastic sampling can be helpful, it is less effective for multiple-choice tasks, where it requires many repetitions of inference and careful temperature tuning. By establishing a baseline for uncertainty-based LMMs ranking, this study aims to motivate and inspire further research into this important area.

**Potential future directions.** Beyond uncertainty scores, several promising directions remain for future research. Test-time augmentation and semantic entropy present natural next steps. Additionally, the internal states of LMMs (Azaria & Mitchell, 2023; Gottesman & Geva, 2024) offer opportunities for uncertainty estimation or error detection, for example, by training a classifier on these representations. However, in unsupervised LMM ranking, this approach requires training a separate classifier per model, which can be computationally expensive. An alternative is to measure the statistical distance between a model's internal state for a given response and the distribution of internal states from multiple inference passes. A larger average distance may signal greater uncertainty and potentially indicate a lower model rank. Exploring how internal states can inform LMM ranking is an open direction. Finally, the observed asymmetry in the impact of text and image feature dissimilarity on cross-dataset correlations highlights a valuable area for improving multimodal benchmark design.

## Acknowledgments

We thank all anonymous reviewers for their constructive feedback, which helped improve the quality of this paper. Tongliang Liu is partially supported by the following Australian Research Council projects: FT220100318, DP220102121, LP220100527, LP220200949, and IC190100031.

## Impact Statement

This paper aims to advance machine learning while acknowledging potential societal impacts. Our findings could be misused: adversarial researchers may train models that consistently get ranked higher than other models, despite performing poorly on data. To mitigate this, incorporating robust calibration methods would be helpful, making model confidence accurately reflects uncertainty. This would help promote safe and responsible deployment of our findings.

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

# A. Appendix

In this supplementary material, we first introduce the experimental details including the models, datasets and compute in Appendix B. Next, we visualize the image and text features using t-SNE plots. Last, we show the full results of the correlation study on all datasets.

# B. Experiment Details

## B.1. Datasets

All experiments are conducted on VLMEvalKit (Duan et al., 2024). We consider 8 datasets and their corresponding links to download TSV files via the toolkit:

ScienceQA (Lu et al., 2022) (`https://opencompass.openxlab.space/utils/VLMEval/ScienceQA_TEST.tsv`);

AI2D (Hiippala et al., 2021) (`https://opencompass.openxlab.space/utils/VLMEval/AI2D_TEST.tsv`);

ChartQA (Masry et al., 2022) (`https://opencompass.openxlab.space/utils/VLMEval/ChartQA_TEST.tsv`);

OCRVQA (Mishra et al., 2019) (`https://opencompass.openxlab.space/utils/VLMEval/OCRVQA_TESTCORE.tsv`);

TextVQA (Singh et al., 2019) (`https://opencompass.openxlab.space/utils/VLMEval/TextVQA_VAL.tsv`);

DocVQA (Mathew et al., 2021) (`https://opencompass.openxlab.space/utils/VLMEval/DocVQA_VAL.tsv`);

RealWorldQA (x.ai, 2024) (`https://opencompass.openxlab.space/utils/VLMEval/RealWorld.tsv`);

MMMU (Yue et al., 2024) (`https://opencompass.openxlab.space/utils/VLMEval/MMMU_DEV_VAL.tsv`)

GQA (Ainslie et al., 2023) (`https://opencompass.openxlab.space/utils/VLMEval/GQA_TestDev_Balanced.tsv`)

## B.2. Models

We include a diverse array of large multimodal models from 12 series:

> **LLaVA-V1.5**
>
> ```
> llava_v1.5_7b
> llava_v1.5_13b
> ```

> **LLaVA-NeXT**
>
> ```
> llava_next_mistral_7b
> llava_next_vicuna_7b
> llava_next_vicuna_13b
> ```

> **LLaVA-NeXT-Interleave**
>
> ```
> llava_next_interleave_7b
> llava_next_interleave_7b_dpo
> ```

## LLaVA-OneVision

```
llava_onevision_qwen2_0.5b_ov
llava_onevision_qwen2_7b_ov
llava_onevision_qwen2_0.5b_si
llava_onevision_qwen2_7b_si
```

## ShareGPT4V

```
sharegpt4v_7b
sharegpt4v_13b
```

## InstructBLIP

```
InstructBLIP_7b
InstructBLIP_13b
```

## Eagle

```
Eagle-X5-7B
Eagle-X5-13B
Eagle-X5-13B-Chat
```

## InternVL

```
Mini-InternVL-Chat-2B-V1-5
Mini-InternVL-Chat-4B-V1-5
InternVL2-1B
InternVL2-2B
InternVL2-4B
InternVL2-8B
```

## PaliGemma

```
paligemma-3b-mix-224
paligemma-3b-mix-448
```

## Mantis

```
Mantis-8B-Idefics2
Mantis-8B-clip-llama3
Mantis-8B-siglip-llama3
```

## mPLUG-Owl2

```
mPLUG-Owl2
```

## Qwen2-VL

```
Qwen2-VL-2B-Instruct
```

> **Qwen-VL**
>
> ```
> deepseek_vl2_tiny
> ```

> **LLaVA Prismatic**
>
> ```
> reproduction-llava-v15+7b
> one-stage+7b
> full-ft-multi-stage+7b
> full-ft-one-stage+7b
> in1k-224px+7b
> dinov2-224px+7b
> clip-224px+7b
> siglip-224px+7b
> clip-336px-resize-crop+7b
> clip-336px-resize-naive+7b
> siglip-384px-letterbox+7b
> llama2-no-cotraining+7b
> llava-lvis4v+7b
> llava-lrv+7b
> llava-lvis4v-lrv+7b
> ```

### B.3. Compute and Library

PyTorch version is 2.01.0+cu117. All experiment is run on four A6000 GPUs. All 45 models can be downloaded via Huggingface with different versions of transformer library:

`transformers==4.33.0` for mPLUG-Owl2 (Li et al., 2022) and InstructBLIP (Dai et al., 2023);

`transformers==4.37.0` for LLaVA-V1.5 (Liu et al., 2024a), ShareGPT4V (Chen et al., 2023a), InternVL (Chen et al., 2024) series;

`transformers==latest` for LLaVA-NeXT (Liu et al., 2024b), LLaVA-OneVision (Li et al., 2024a), LLaVA-NeXT-Interleave (Li et al., 2024b) PaliGemma-3B (Beyer et al., 2024), Mantis (Jiang et al., 2024), Eagle (Shi et al., 2024b) and LLaVA Prismatic (Karamcheti et al., 2024) series.

## C. Visualization of Image and Text Features

In the main paper, we demonstrate that distances in text features contribute more significantly to the weak correlation of model performance across datasets than image features. To visualize this finding, we utilize t-distributed stochastic neighbor embedding (t-SNE) (Hinton & Roweis, 2002). Our results show that CLIP-L-14-336 (Radford et al., 2021) effectively captures distinctions between images from different datasets, with images from the same dataset forming distinct clusters. Additionally, the text features of visual question answering (VQA) and multiple-choice visual questioning (MCVQ) tasks are separated by the text encoder of CLIP. Notably, the text features of OCRVQA are distant from those of ChartQA, TextVQA, and DocVQA, despite all being VQA tasks. This finding supports our observation in the main paper that the effectiveness of ranking methods is influenced by dataset characteristics (*i.e.*, VQA *vs.* MCVQ).

## D. Full Results of Correlation Study

In the following, we present the full results of correlation study for ranking different series of large multimodal models. We only show $\textbf{NLL}_\text{F}$, $\textbf{NLL}_\text{P}$, $\textbf{NLL}_\text{min}$, $\textbf{NLL}_\text{mean}$, $\textbf{Ent}_\text{F}$, $\textbf{Ent}_\text{P}$, $\textbf{Ent}_\text{max}$, $\textbf{Ent}_\text{mean}$, $\textbf{Sample}_\text{BLEU}$, $\textbf{Sample}_\text{BERT}$, $\textbf{Sample}^*_\text{BERT}$. ATC and Accuracy on the line are not included because their performance are computed by the average correlation strength across eight datasets.

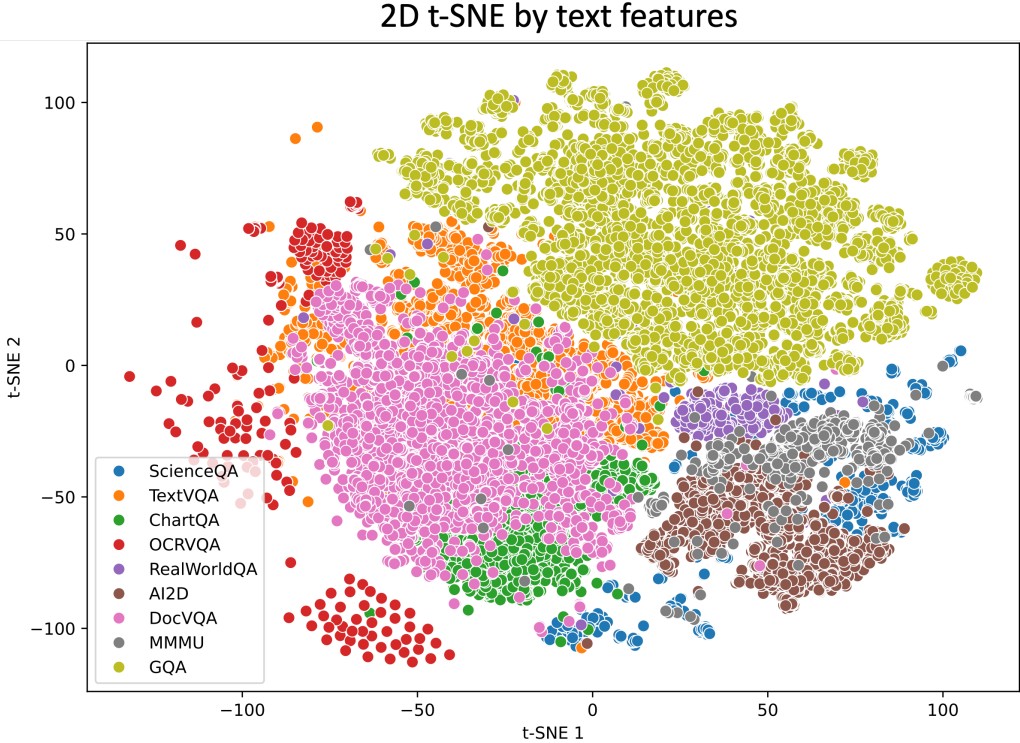

*Figure 5.* **Two t-SNE plots are presented: one using image features (Top) and the other using text features (Bottom) of the datasets.** We observe that the text features of OCRVQA are more scattered and significantly distant from those of other datasets.

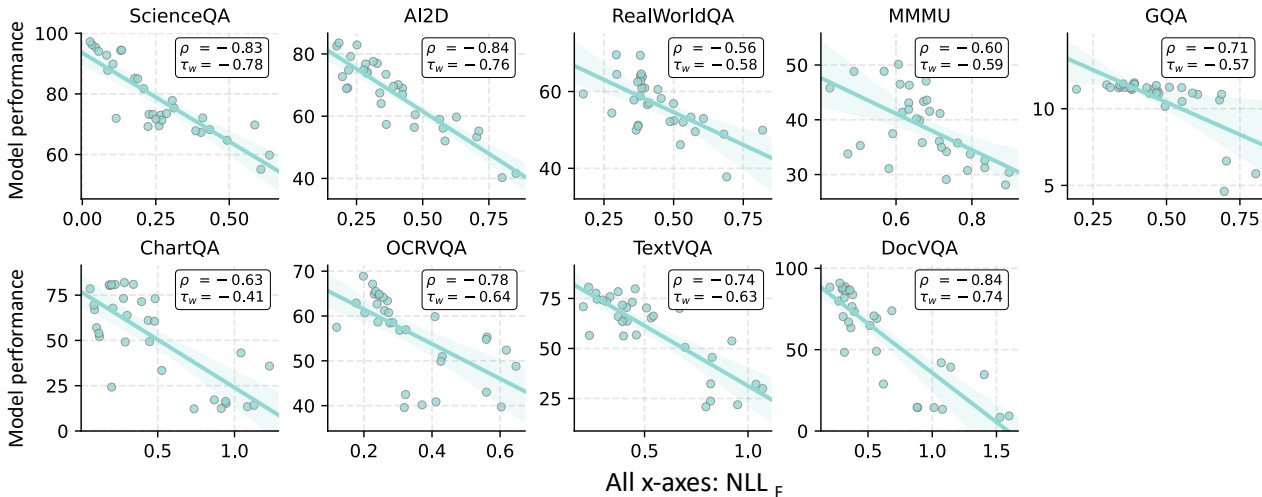

*Figure 6.* **Correlation study between NLL$_F$ and model performance.** Spearman's correlation ($\rho$) and weighted Kendall's correlation ($\tau_w$) are metrics. Each point denotes a model. Straight lines are fit with robust linear regression (Huber, 2011).

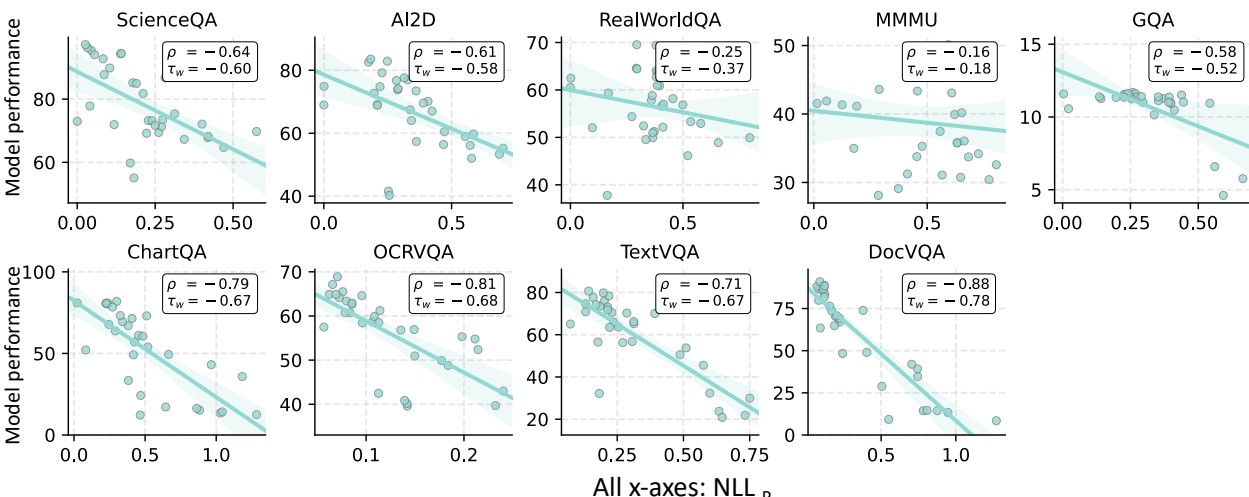

*Figure 7.* **Correlation study between NLL$_P$ and model performance.** Spearman's correlation ($\rho$) and weighted Kendall's correlation ($\tau_w$) are metrics. Each point denotes a model. Straight lines are fit with robust linear regression (Huber, 2011).

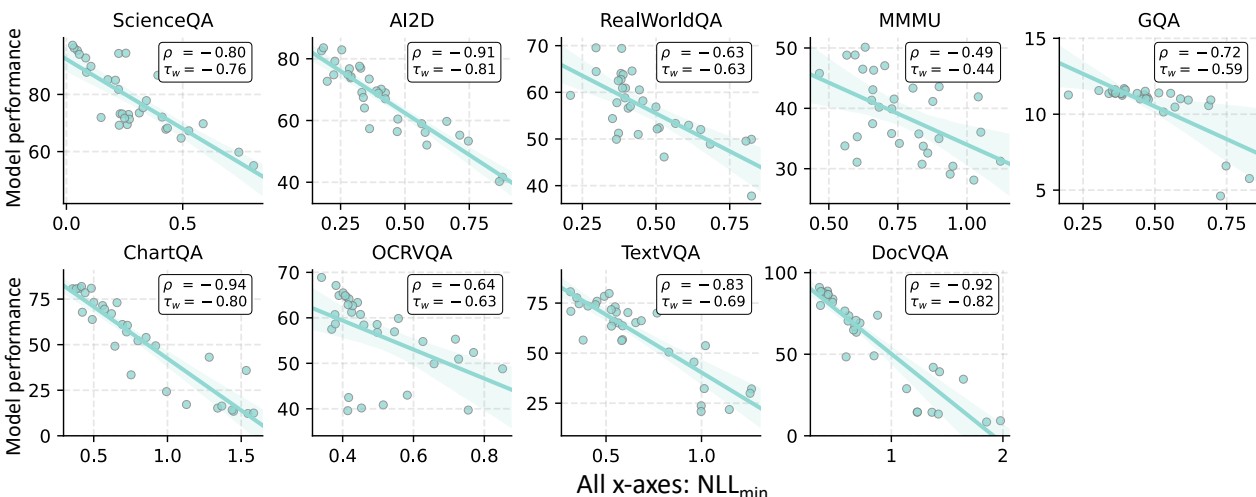

*Figure 8.* **Correlation study between NLL_min and model performance.** Spearman's correlation ($\rho$) and weighted Kendall's correlation ($\tau_w$) are metrics. Each point denotes a model. Straight lines are fit with robust linear regression (Huber, 2011).

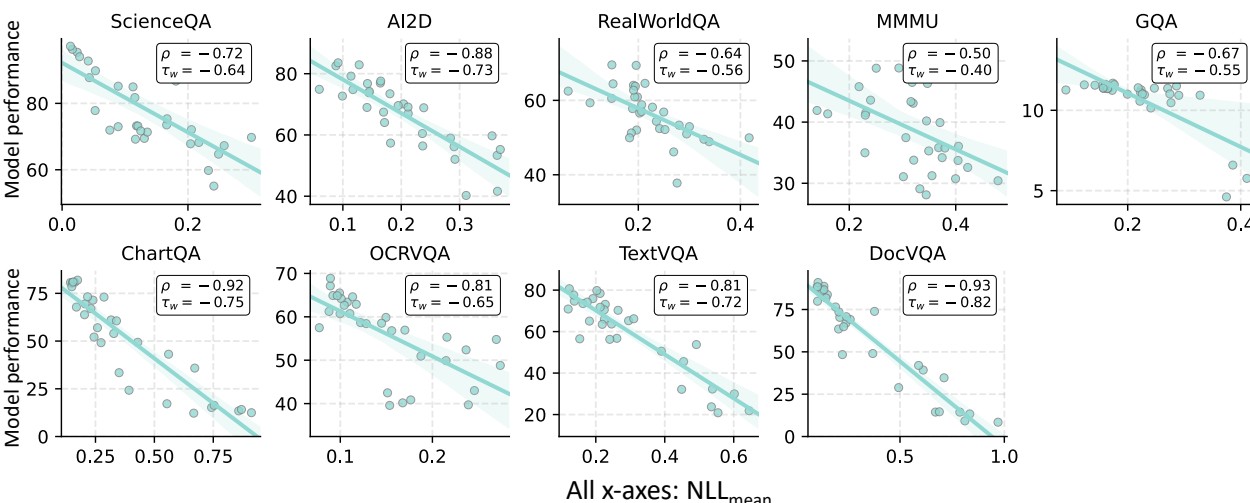

*Figure 9.* **Correlation study between NLL_mean and model performance.** Spearman's correlation ($\rho$) and weighted Kendall's correlation ($\tau_w$) are metrics. Each point denotes a model. Straight lines are fit with robust linear regression (Huber, 2011).

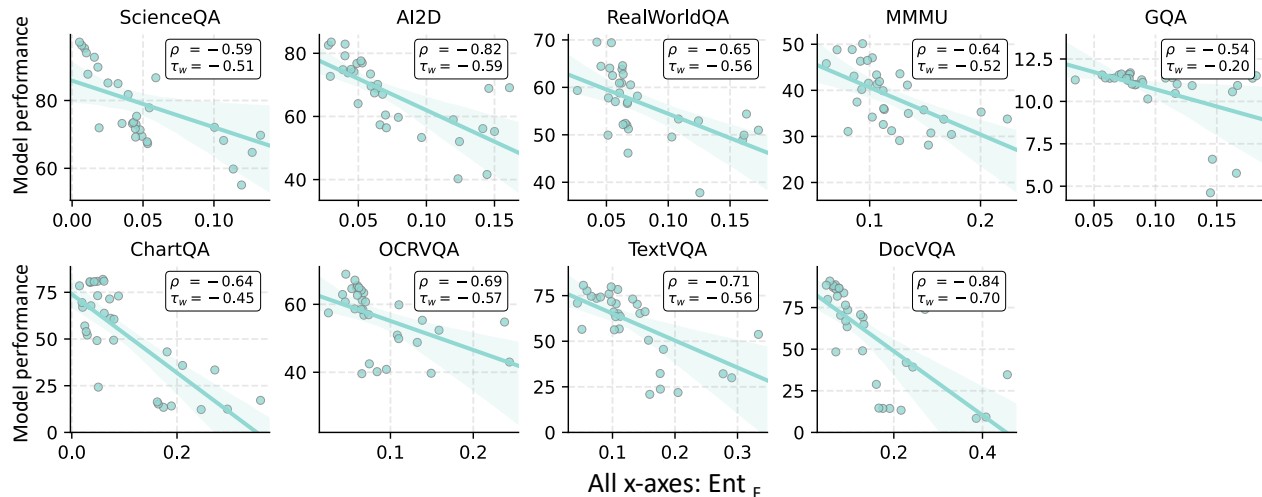

*Figure 10.* **Correlation study between Ent$_F$ and model performance.** Spearman's correlation ($\rho$) and weighted Kendall's correlation ($\tau_w$) are metrics. Each point denotes a model. Straight lines are fit with robust linear regression (Huber, 2011).

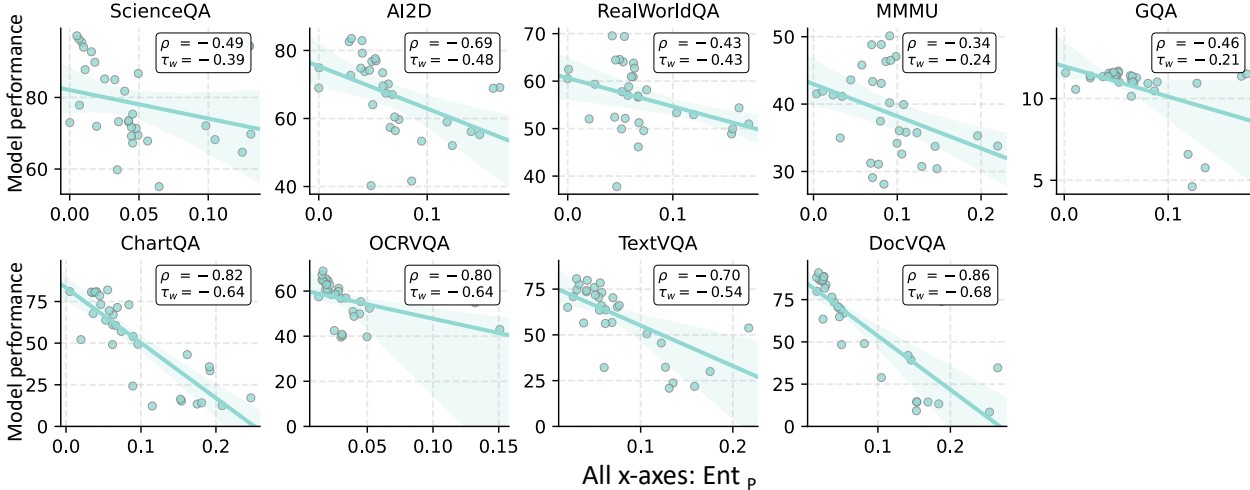

*Figure 11.* **Correlation study between Ent$_P$ and model performance.** Spearman's correlation ($\rho$) and weighted Kendall's correlation ($\tau_w$) are metrics. Each point denotes a model. Straight lines are fit with robust linear regression (Huber, 2011).

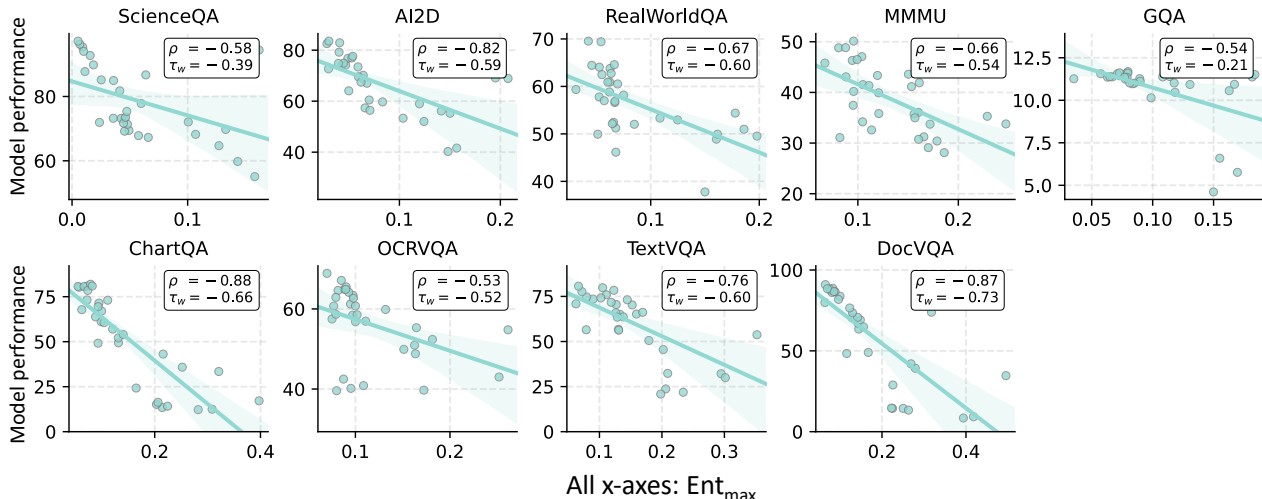

*Figure 12.* **Correlation study between Ent$_{max}$ and model performance.** Spearman's correlation ($\rho$) and weighted Kendall's correlation ($\tau_w$) are metrics. Each point denotes a model. Straight lines are fit with robust linear regression (Huber, 2011).

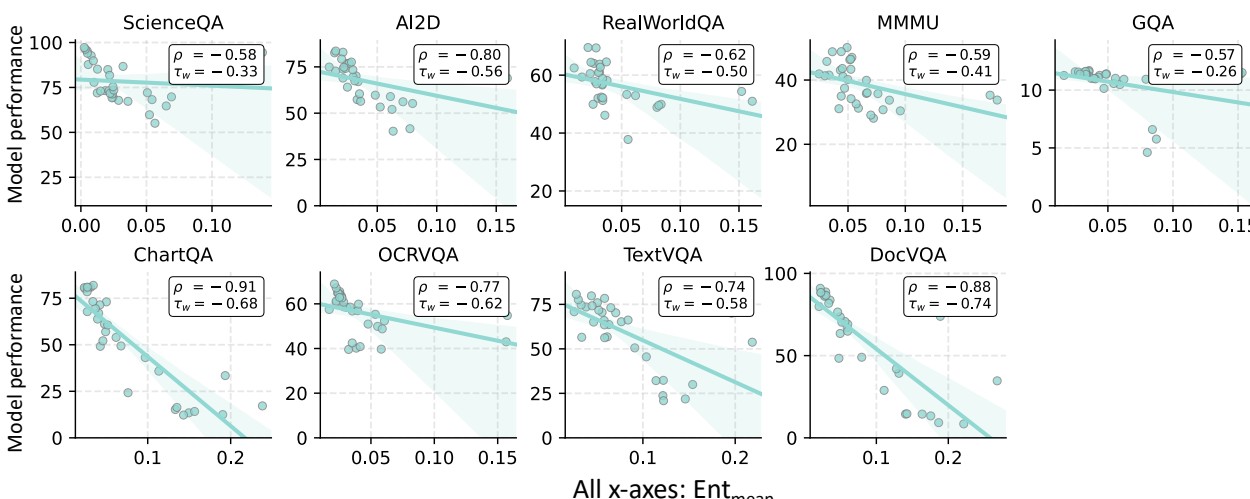

*Figure 13.* **Correlation study between Ent$_{mean}$ and model performance.** Spearman's correlation ($\rho$) and weighted Kendall's correlation ($\tau_w$) are metrics. Each point denotes a model. Straight lines are fit with robust linear regression (Huber, 2011).

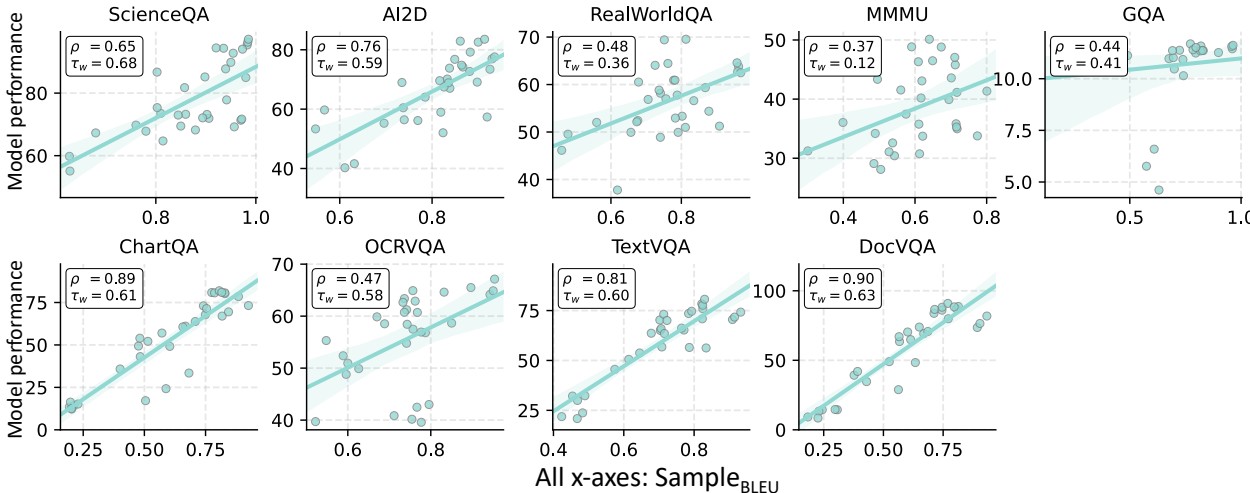

Figure 14. **Correlation study between Sample$_{\text{BLEU}}$ and model performance.** Spearman's correlation ($\rho$) and weighted Kendall's correlation ($\tau_w$) are metrics. Each point denotes a model. Straight lines are fit with robust linear regression (Huber, 2011).

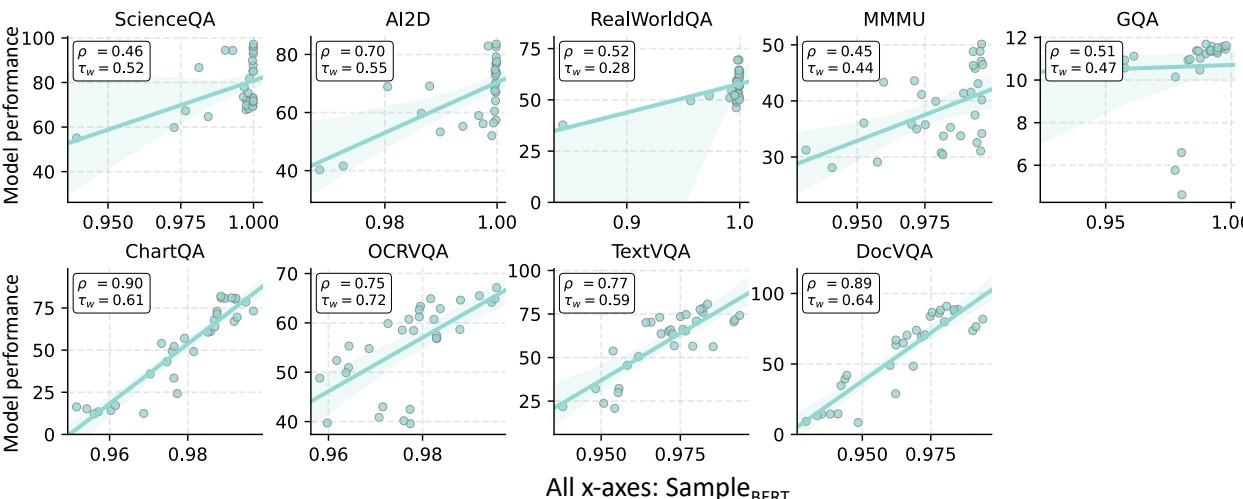

Figure 15. **Correlation study between Sample$_{\text{BERT}}$ and model performance.** Spearman's correlation ($\rho$) and weighted Kendall's correlation ($\tau_w$) are metrics. Each point denotes a model. Straight lines are fit with robust linear regression (Huber, 2011).

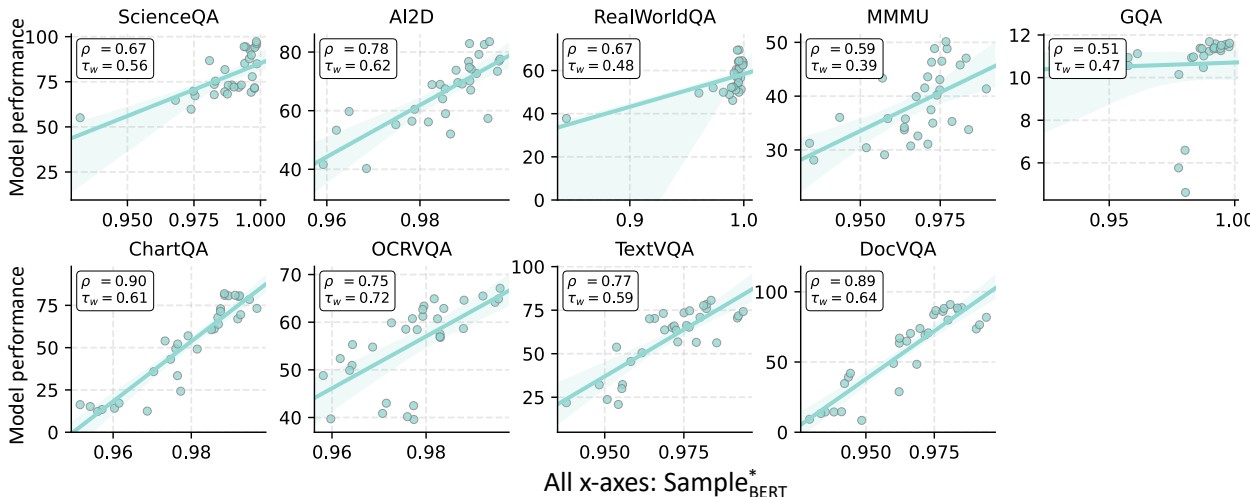

*Figure 16.* **Correlation study between Sample*$^*_{BERT}$* and model performance.** Spearman's correlation ($\rho$) and weighted Kendall's correlation ($\tau_w$) are metrics. Each point denotes a model. Straight lines are fit with robust linear regression (Huber, 2011).

