# OpenReview forum: "Ranked from Within: Ranking Large Multimodal Models Without Labels"
_ICML.cc/2025/Conference — ICML 2025 poster_

### Official Review · Reviewer_Fjnt · 2025-03-11

**Overall Recommendation:** 2

**Summary:**

This paper addresses the challenge of ranking large multimodal models (LMMs) without access to labeled data. Specifically, the authors propose uncertainty-based ranking methods that utilize softmax probabilities, self-consistency, and labeled proxy sets to estimate model performance. They evaluate 45 LMMs on eight benchmarks and find that uncertainty-based metrics, particularly negative log-likelihood (NLL), provide a robust ranking mechanism. The study suggests that relying solely on benchmark performance for ranking models may be unreliable due to domain shifts.

**Claims And Evidence:**

Yes. The study tests 45 LMMs across eight diverse benchmarks, demonstrating strong empirical rigour.

**Essential References Not Discussed:**

This paper contains enough references.

**Ethics Expertise Needed:**

["Other expertise"]

**Experimental Designs Or Analyses:**

Yes. The analysis encompasses multiple ranking methods, ablation studies, and cross-domain correlation assessments, ensuring a comprehensive evaluation.

**Methods And Evaluation Criteria:**

The analysis encompasses multiple ranking methods, ablation studies, and cross-domain correlation assessments, ensuring a comprehensive evaluation. However, the authors do not provide clear guidance on selecting appropriate ranking methods for different scenarios.

**Other Comments Or Suggestions:**

Please refer to the weaknesses.

**Other Strengths And Weaknesses:**

Weaknesses:
1. Although the authors conduct many experiments testing the model performance across many ranking methods(NLL loss, Entropy, BLEU, and BERTScores), they do not provide clear guidance on selecting appropriate ranking methods for different scenarios.
2. In Models of Section 5.1, the authors claim to provide two settings. However, they only explain one, while the other setting is missing.
3. The novelty of this paper seems limited. The authors only test many existing ranking methods (NLL loss, Entropy, BLEU, and BERTScores) without providing new insights.
4. In Section 6, the authors only evaluate the series of LLaVA models, raising concerns about the generality of the analysis.
5. In Section 5.1, some citations appear as question marks, indicating missing or improperly formatted references.

**Questions For Authors:**

Please refer to the weaknesses.

**Relation To Broader Scientific Literature:**

This paper estimates model performance using ranking methods, such as NLL loss, Entropy, BLEU, and BERTScores.

**Theoretical Claims:**

N/A. This paper does not include any theoretical claims.

---

> ### Author Rebuttal · Authors · 2025-03-31
>
> > Q1. The novelty of this paper seems limited. The authors only test many existing ranking methods (NLL loss, Entropy, BLEU, and BERTScores) without providing new insights.
>
> **A**: While NLL, Entropy, BLEU, and BERTScore have been widely used for uncertainty quantification in LLMs, to the best of our knowledge they have not been explored for unsupervised LMM ranking. Our work is the first to systematically investigate their applicability in this context. Through extensive experiments across diverse models and benchmarks, we demonstrate that $NLL_{max}$ is the most reliable and accurate ranking proxy measure. Additionally, we introduce a simple yet effective modification (*i.e.*, $Sample_{BERT}^*$) to $Sample_{BERT}$, improving its consistency. Furthermore, we observe that self-consistency methods perform better for LMM ranking in VQA than MCQA, and that $Sample_{BERT}^*$ outperforms BLEU.
>
> In the revised version, we will highlight our core findings: (1) model performance on one dataset does not reliably reflect its ranking on another; (2) uncertainty in model predictions can be predictive of model rank, **with $NLL_{max}$ being the most accurate technique in general**; and (3) text prompt similarity better correlate with model performance across datasets than image feature similarity.
>
> > Q2. They do not provide clear guidance on selecting appropriate ranking methods for different scenarios.
>
> **A**: We agree this would strengthen the paper. In the revision, we provide guidance on selecting appropriate ranking methods based on model accessibility and computational constraints, as follows.
>   - $NLL_{max}$ is generally the most accurate, efficient and consistent method for LMM performance ranking.
>   - While self-consistency methods are also competitive for VQA, they require K-times the compute, where K is the number of unique inference outputs per prompt. However, for closed-source models where internal statistics (*e.g*., logits) are unavailable, self-consistency-based methods are more suitable. Among these, $Sample_{BERT}^*$ is the recommended approach.
>
> > Q3. In Models of Section 5.1, the authors claim to provide two settings. However, they only explain one, while the other setting is missing.
>
> **A**: Thank you for picking up this error - it is a typo. Section 5 only contains one setting, where different series of models are evaluated. We have corrected this in the revision.
>
> > Q4. In Section 6, the authors only evaluate the series of LLaVA models, raising concerns about the generality of the analysis.
>
> **A**: Section 6 includes three distinct analyses. The first specifically investigates whether the considered methods are effective for ranking models within the same series. We use LLaVA models as a case study because they are widely adopted and representative of LMM architectures. Furthermore, the diversity within the LLaVA series is ensured by incorporating LLaVA-prismatic models trained with varying visual encoders and large language models, making the analysis more comprehensive. We will clarify this in the paper.
>
> The other two analyses in Section 6 consider models from different series, consistent with the broader evaluation in Section 5. We have rewritten this section to make this clearer.
>
> > Q5. In Section 5.1, some citations appear as question marks, indicating missing or improperly formatted references.
>
> **A**: Thank you for your meticulous comment. we will double-check and correct the improperly formatted references.

---

### Official Review · Reviewer_isqi · 2025-03-12

**Overall Recommendation:** 2

**Summary:**

This paper presents a study on ranking methods for evaluating large language models (LLMs) without accessing labels. The authors main findings are that Accuracy-on-the-Line is unreliable for ranking LLMs in new domains, the choice of token significantly impacts output probability-based ranking, and the negative log-likelihood (NLL) is more stable and predictive for LLM ranking than other evaluation methods.

**Claims And Evidence:**

Yes, the claims are well-supported and make sense in light of the experimental results.

**Essential References Not Discussed:**

In the section "Related Work: Evaluation & Benchmarking LMMs," the paper mentions RealWorldQA and several other benchmarks for evaluation purposes. However, it fails to address some commonly used benchmarks for LLMs, such as CV-benchmarks (e.g., GQA, MMVP, OCRBench) and math-benchmarks (e.g., MathVerse, MathVista).

**Experimental Designs Or Analyses:**

Yes, the experimental designs and analyses appear to be sound based on the reported correlation studies.

**Methods And Evaluation Criteria:**

Yes, this manuscript introduces a ranking method that effectively evaluates LLMs without relying on labeled data. The NLL (Negative Log-Likelihood) metric demonstrates highly stable performance indicators for target domains.

**Other Comments Or Suggestions:**

To enhance the comprehensiveness of the evaluation, the authors should
1.consider incorporating more vision and spatial understanding-related benchmarks, such as GQA and MMVP.
2.They should also update their model selection to include recently developed LLMs like the Qwen series and DeepSeek series to validate the universality of their proposed criteria.
3.Finally, the authors should elaborate on the applicability of their evaluation framework to closed-source models to ensure its relevance across different model types.

**Other Strengths And Weaknesses:**

Strengths:
The authors have investigated the important issue of evaluating models using unlabeled data and proposed a corresponding ranking evaluation method. The results make sense and have yielded many findings beneficial to the development of the community. The authors evaluated the performance on 45 LLMs, demonstrating good breadth.
Weaknesses:
However,
1.the benchmarks used by the authors for evaluation seem limited and unbalanced. The paper employs four types of benchmarks: Optical Character Recognition, Science, Vision, and General. But the authors appear to focus mainly on document OCR-related benchmarks (such as TextVQA, ChartVQA, DocVQA, OCRVQA), with very little attention to vision-related benchmarks (only one RWQA). There seems to be a lack of evaluation in areas that are also very important for LLM assessment, such as the evaluation of LLMs' spatial understanding capabilities.
2.In addition, the evaluation models used by the authors also seem limited, involving only some LLaVA series models or relatively older models like InstructBLIP and mPLUG-Owl2. There is a lack of assessment of newly developed models, such as the Qwen series and DeepSeek series. It needs further examination whether these models still follow the evaluation criteria proposed by the authors. The authors should also clarify whether the evaluation of closed-source models follows the proposed pattern.

**Questions For Authors:**

Please see weakness and suggestions above.

**Relation To Broader Scientific Literature:**

The paper points out the unreliability of evaluating and ranking LLMs across different datasets, and provides a method for assessing the ranking of LMMs in the absence of labeled data. This method yields more stable evaluation results across different domains compared to previous approaches. It holds significant importance for further research into the performance of LLMs and issues related to their deployment.

**Theoretical Claims:**

This work does not include proofs for its theoretical claims. However, based on the results provided by the authors, the findings appear to be reasonable.

---

> ### Author Rebuttal · Authors · 2025-03-31
>
> > Q1. In the section "Related Work: Evaluation & Benchmarking LMMs," the paper mentions RealWorldQA and several other benchmarks for evaluation purposes. However, it fails to address some commonly used benchmarks for LLMs, such as CV-benchmarks (e.g., GQA, MMVP, OCRBench) and math-benchmarks (e.g., MathVerse, MathVista).
>
> **A**: Thank you for pointing out these benchmarks. We will include them in the related work. The following text will be used in Section 2:
>
> *“Several benchmarks (x.ai, 2024; Ainslie et al., 2023; Tong et al., 2024) have been developed to assess multimodal models' real-world spatial understanding. Lu et al. (2024) and Zhang et al. (2024) introduce benchmarks specifically designed to evaluate MLLMs' mathematical reasoning, focusing on their ability to comprehend and reason visual mathematical figures. Additionally, numerous benchmarks (Mishra et al., 2019; Masry et al., 2022; Mathew et al., 2021; Liu et al., 2024) assess the performance of LMMs in optical character recognition.”*
>
>  *[1] x.ai. Grok 1.5v: The future of ai models, 2024. URL https://x.ai/blog/grok-1.5v.*
>
>  *[2] Ainslie, J., et al. Gqa: Training generalized multi-query transformer models from multi-head checkpoints. In EMNLP, 2023.*
>
>  *[3] Tong, S., et al. Eyes wide shut? exploring the visual shortcomings of multimodal llms. In CVPR, 2024.*
>
>  *[4] Lu, P., et al. Mathvista: Evaluating mathematical reasoning of foundation models in visual contexts. In ICLR, 2024.*
>
>  *[5] Zhang, R., et al. Mathverse: Does your multi-modal llm truly see the diagrams in visual math problems? In ECCV, 2024.*
>
>  *[6] Liu et al., Ocrbench: on the hidden mystery of ocr in large multi-modal models. Science China Information Sciences, 67(12), December 2024d.*
>
> > Q2. Consider incorporating more vision and spatial understanding-related benchmarks, such as GQA and MMVP.  They should also update their model selection to include recently developed LLMs like the Qwen series and DeepSeek series to validate the universality of their proposed criteria.
>
> **A**: Thank you for your valuable suggestion. Following your recommendation, we incorporated four new models—QwenVL, QwenVL2, DeepSeek-VL, and DeepSeek-VL2—and added GQA as an additional dataset for evaluation. We summarize the results in the table below, where the metric is rank correlation, and AoL performance represents the average correlation strength when using the other eight domains to predict rankings in the target domain.
>
> - Generalizability to new models: The uncertainty-based ranking methods (*e.g.*, $NLL_{max}$) successfully extend to newly introduced models.
>
> - Performance on new dataset GQA: We observe consistent trends: (1) AoL does not reliably rank model performance, and (2) $NLL_{max}$ provides a more stable and indicative ranking signal compared to other methods.
>
> These new results further support the applicability and robustness of the evaluated uncertainty-based methods. We will incorporate them into the revised version.
>
> | |  AoL  | $NLL_{F}$ | $NLL_{P}$ | $NLL_{max}$ | $NLL_{mean}$ | $Ent_{F}$ | $Ent_{P}$ | $Ent_{max}$ | $Ent_{mean}$ | $Sample_{BLEU}$ | $Sample_{BERT}$ | $Sample_{BERT}^*$ |
> |--------------|:----:|:-----:|:-----:|:-------:|:--------:|:-----:|:-----:|:-------:|:--------:|:-----------:|:-----------:|:------------:|
> | **SQA-I**   | 0.54 | 0.84  | 0.66  | **0.81**    | 0.74     | 0.63  | 0.50  | 0.62    | 0.59     | 0.68        | 0.51        | 0.70         |
> | **ChartQA** | 0.68 | 0.62  | 0.79  | **0.94**    | 0.92     | 0.64  | 0.82  | 0.89    | 0.91     | 0.86        |  0.87     | 0.87         |
> | **GQA**     | 0.59 | 0.64  | 0.55  | **0.70**   | 0.63     | 0.47  | 0.46  | 0.52    | 0.52     |   0.59    |    0.61      |  0.61      |
>
>
> > Q3. Finally, the authors should elaborate on the applicability of their evaluation framework to closed-source models to ensure its relevance across different model types.
>
> **A**: We appreciate the reviewer’s insightful suggestion. While our experiments focus on models with accessible internal outputs, we note that **our evaluation framework can be extended to closed-source models as well**—particularly through methods that rely solely on final predictions, such as $Sample_{BERT}$. These approaches operate without access to logits or internal statistics and can support **label-free model selection** even in restricted-access settings.
>
> We agree this distinction is important and will clarify in the manuscript that our framework can be **applied to closed-source models**, depending on the availability of output-level information, without requiring structural modifications or access to internals.

---

### Official Review · Reviewer_V6Xj · 2025-03-12

**Overall Recommendation:** 3

**Summary:**

The paper investigates alternative ways to rank large multimodal models on new domains in absence of ground truth annotations. The authors compare 3 types of approaches: (1) labeled proxy datasets (AoL: where the performance on N-1 datasets is used to predict the performance on the Nth dataset; ATC: where proxy datasets are used to extract a confidence threshold to be used for the new unlabeled dataset), probability-based (NLL, Entropy), and self-consistency (sample-based).  By running an extensive analysis with 45 models (30 different models and their versions) on 8 multiple-choice or open-ended visual QAs, the authors conclude that probability-based metrics tend to be more reliable and show strong correlations with actual performance, compared to AoL or ATC. An additional analysis is provided for AoL, to shed light on its low correlation with performance, showing that text similarities across datasets is more indicative of strong correlation with performance than image similarities.

## update after rebuttal ##
I read the rebuttal and the other reviews and I maintain my initial score.

**Claims And Evidence:**

This is a compelling submission, with potential for impact in how benchmarking of large multimodal models is approached.
The claims are clearly presented and backed by an extensive set of experiments.

**Essential References Not Discussed:**

These works should be discussed as well

The Internal State of an LLM Knows When It’s Lying https://aclanthology.org/2023.findings-emnlp.68.pdf
Estimating Knowledge in Large Language Models Without Generating a Single Token https://aclanthology.org/2024.emnlp-main.232.pdf

**Experimental Designs Or Analyses:**

The empirical analysis is extensive and covers well the space of multimodal models.

**Methods And Evaluation Criteria:**

The proposed analysis is comprehensive and robust, including multiple metrics (probabilistic, sample-based, proxy-labelled).

**Other Comments Or Suggestions:**

The paper is well written.
A few typos:
L295: Prob_min, Prob_avg are not used in the table, they should be NLL_max, NLL_avg?
L247-248, some citations are incorrect and appear as “?”

**Other Strengths And Weaknesses:**

The impact section could include a statement about the possibility of training models to elicit high confidence when producing deceiving answers.

**Questions For Authors:**

1. Would it be possible to train models to elicit high confidence when producing deceiving answers?

**Relation To Broader Scientific Literature:**

The related work section covers relevant works, but some are missing, see below.

**Theoretical Claims:**

NA

---

> ### Author Rebuttal · Authors · 2025-03-31
>
> > Q1. These works should be discussed as well: (1) The Internal State of an LLM Knows When It’s Lying  (2) Estimating Knowledge in Large Language Models Without Generating a Single Token
>
> **A**: Thank you for your valuable suggestions. We will include these two studies, along with other relevant works on the internal states of LLMs and LMMs, in the related work section. Additionally, we will discuss the limitations of using internal states for model ranking and potential directions for future research as below:
>
> *"Internal states of LMMs can be leveraged for uncertainty quantification or error detection by training a classifier. However, in the context of unsupervised LMM ranking, a separate classifier must be trained for each LMM, making the approach computationally expensive. One potential alternative is to measure the statistical distance between the internal state of the original answer and the distribution of internal states of multiple inference processes. A larger average distance may indicate higher uncertainty in model predictions, suggesting a lower model rank. It would be interesting to investigate the use of internal states for LMM ranking is left as future work.”*
>
> > Q2. The impact section could include a statement about the possibility of training models to elicit high confidence when producing deceiving answers.
>
> **A**: Thank you for the suggestion. We will incorporate this discussion into the broader impact section:
>
> *"Our findings could be misused: adversarial researchers may train models that consistently get ranked higher than other models, despite performing poorly or adversarially on data. To mitigate this risk, incorporating robust calibration methods is helpful, ensuring that model confidence accurately reflects uncertainty. This would help promote safer and more responsible deployment of our findings."*
>
> > Q3. A few typos: L295: $Prob_{min}$, $Prob_{avg}$ are not used in the table, they should be $NLL_{max}$, $NLL_{avg}$? L247-248, some citations are incorrect and appear as “?”
>
> **A**: Thank you for your meticulous comment. The $Prob_{min}$ and $Prob_{avg}$ should be $NLL_{max}$ and $NLL_{avg}$. We will carefully review the manuscripts and correct all typos.

---

### Official Review · Reviewer_HFDC · 2025-03-14

**Overall Recommendation:** 3

**Summary:**

This paper explores how to effectively evaluate the performance of LMMs without requiring task-specific labels, and systematically validates three types of model uncertainty-based approaches, including softmax probabilities, self-consistency, and labeled proxy sets.
Comprehensive experiments across various LMMs and benchmarks show that uncertainty scores derived from softmax distributions provide a robust and consistent basis for ranking models across various tasks, facilitating the ranking of LMMs on unlabeled data.

### update after rebuttal
The author's rebuttal addressed several of my concerns. I keep my original rating.

**Claims And Evidence:**

Strengths:
- This paper presents an important and interesting problem regarding how to evaluate LMMs without requiring task-specific labels.
- The authors perform extensive experiments to analyze how well uncertainty-based metrics can measure relative model performance, resulting in some empirical findings.

Weaknesses:
- Although the authors derived a series of empirical conclusions through extensive experimental analysis,
    - 1) they did not propose new methods or metrics based on these findings for selecting the optimal LMM for a given task in an unlabeled scenario.
    - 2) The findings in this paper lack universality, are somewhat diffuse, and would benefit from a more concise formulation. As a reader, these findings cannot directly guide me to quickly and effectively select the optimal LMM for a given task without manual labels.

**Essential References Not Discussed:**

No

**Experimental Designs Or Analyses:**

Yes.

**Methods And Evaluation Criteria:**

Yes.

**Other Comments Or Suggestions:**

- making the findings more concise and instructive.

**Other Strengths And Weaknesses:**

Other Strengths:
- The paper additionally provides detailed correlation analysis of model performance.

**Questions For Authors:**

- What is the core finding of this paper?
- How does this finding contribute to the LMM community?
- How can the readers employ this finding to evaluate LMMs without label annotations?

**Relation To Broader Scientific Literature:**

While the paper focuses on an intriguing and valuable problem and conducts extensive experimental analysis, its contribution and impact on the broader literature are not fully clear. Because the authors did not introduce novel methods and all implemented uncertainty-based methods are pre-existing.

**Theoretical Claims:**

This paper does not involve the claim and proof of novel theories.

---

> ### Author Rebuttal · Authors · 2025-03-31
>
> > Q1. Authors did not propose new methods or metrics based on these findings for selecting the optimal LMM for a given task in an unlabeled scenario.
>
> **A**: We acknowledge that our work does not introduce new methods or metrics. However, it addresses a previously under-explored yet practically important problem: how to evaluate and select LMMs in unlabeled scenarios.  We systematically analyze existing uncertainty quantification techniques across diverse models and datasets, offering insights into their effectiveness and failure cases. These indicators serve as practical tools for model selection without labels, which we believe is a critical step toward reliable LMM evaluation. Our findings also highlight the promise of uncertainty-based approaches, motivating future research into new metrics for this setting.
>
> > Q2. What is the core finding of this paper?
>
> **A**: Our key findings are: (1) model performance on one dataset does not reliably predict its ranking on another; (2) uncertainty in model predictions can be predictive of model rank, **with $NLL_{max}$ being the most accurate technique in general**; and (3) text prompt similarity better correlate with model performance across datasets than image feature similarity. We have clarified this in the revised paper, making these observations more crisply than in the initial submission.
>
> > Q3. How does this finding contribute to the LMM community?
>
> **A:** Our findings highlight three key contributions to the LMM community. First, the weak correlation of model performance across datasets underscores the need for diverse multimodal benchmarks to comprehensively assess LMM capabilities. Second, we find that text prompt similarity has a stronger influence on model performance correlations across datasets than image feature similarity.  This finding emphasizes the importance of evaluating LMMs with diverse textual prompts rather than focusing solely on image variation. Third, the effectiveness of baseline uncertainty quantification methods suggests uncertainty estimation as a promising approach for unsupervised LMM ranking.
>
> > Q4. How can the readers employ this finding to evaluate LMMs without label annotations?
>
> **A:** Given a test dataset without labels and multiple LMMs to evaluate, our findings provide a practical framework for ranking and selecting the most suitable model. Specifically, there are two approaches:
>   - For **models with access to internal statistics (*e.g*., logits)**, their $NLL_{max}$ scores can be directly used to rank model performance.
>   - For **closed-source models**, where internal statistics are unavailable, $Sample_{BERT}^*$ serves as a viable alternative for effective performance ranking.

---

### Decision · Program_Chairs · 2025-05-01

**Decision:**

Accept (poster)

**Comment:**

## Summary
This paper explores how to effectively evaluate the performance of LMMs without requiring task-specific labels and systematically validates three types of model uncertainty-based approaches, including softmax probabilities, self-consistency, and labeled proxy sets. Comprehensive experiments across various LMMs and benchmarks show that uncertainty scores derived from softmax distributions provide a robust and consistent basis for ranking models across various tasks, facilitating the ranking of LMMs on unlabeled data.

## Reviewer Consensus and Review Quality
The submission received four reviews. Reviewer HFDC and V6Xj provided positive rating, highlighting the good novelty and experiments. Reviewer isqi and Fjnt expressed concerns about the experiment design. In the rebuttal, the authors partially address these concerns by **adding the experiments of QwenVL, DeepSeek-VL, and so on**.

## Strengths
- This paper presents an important and interesting problem regarding how to evaluate LMMs without requiring task-specific labels.
- The authors perform extensive experiments to analyze how well uncertainty-based metrics can measure relative model performance, resulting in some empirical findings.

## Weaknesses
- They did not propose new methods or metrics based on these findings for selecting the optimal LMM for a given task in an unlabeled scenario.
- The findings in this paper lack universality, are somewhat diffuse, and would benefit from a more concise formulation.

## Justification
Based on the reviewers’ comments and the author’s response, I believe this paper has potential but needs more work. While the paper did not present a concise suggestion to select the optimal LLM, the strengths of the paper outweigh its weaknesses. In addition, it ranks 2 among 12 papers. Therefore, I recommend **weak accept**.